# Multi-Scale Window based Transformer Network for High Quality Image Inpainting

## Abstract

To achieve effective image inpainting, it is crucial for the model to understand contextual information. Previous studies using CNN-based algorithms have encountered limitations due to the absence of long-range dependencies, which resulted in the model's inability to capture contextual information. In this paper, we propose a Multi-Scale Window-based Transformer model for high-quality image inpainting. We introduce a transformer network with multi-scale windows to capture the influence of different window sizes and gather significant contextual information. To effectively integrate features processed through self-attention, we modified the polarized self-attention network to align with the dimensions of the multi-window scale. We also propose the Selective Mask Update method, which captures vital information from features processed by self-attention, enabling the generation of higher-quality results. Experiments show that it effectively fills in missing areas and demonstrates superior performance on the benchmark dataset compared to other models.

## 1 Introduction

Image inpainting (completion) aims to restore damaged images by filling in their empty areas with plausible content. This technique can be applied in various areas, including photo editing (Jo & Park, 2019) and image restoration (Wan et al., 2018). It can also be utilized for object removal (Shetty & Schiele, 2018), removing unwanted objects within images.

To achieve successful image inpainting, it is crucial for the model to grasp contextual information. The contextual information refers to the inferred information derived from the surrounding pixels of the missing areas in the image. Sufficient availability of contextual information is essential for creating the shape, structure, and texture of the missing regions. Previous studies (Yan et al., 2022; Yu et al., 2021) have utilized a convolutional neural network (CNN) with an encoder-decoder architecture to comprehend relevant surrounding information for the purpose of restoration. However, these CNN-based algorithms are effective in restoring small missing areas but struggle with larger ones. For images with large missing areas, the spatial distance between the normal regions and the areas that need to be filled increases. This makes it challenging to perceive and effectively utilize meaningful contextual information. This issue arises from the limitation of the receptive field of CNNs, which hinders the learning of global features and long-range dependencies (Yu et al., 2018)

To address this issue, a model based on the Transformer architecture (Vaswani et al., 2017), which incorporates self-attention as a fundamental component in every layer, has been introduced. Image inpainting algorithms based on the Transformer excel in capturing global information and, as a result, they are more capable of utilizing meaningful contextual information compared to algorithms relying on CNNs. For these reasons, recent advancements in image inpainting have led to the proposal of Transformer-based models (Yan et al., 2018; Zhang et al., 2018a).

Most models based on Vision Transformers (ViT) (Dosovitskiy et al., 2021) divide input images into tokens of a single patch size for self-attention or use tokens divided into windows for self-attention within each window, as seen in the Swin-Transformer (Liu et al., 2021b) based models. It often uses a fixed window size per transformer block for conducting self-attention. This approach fails to account for the impact of different window sizes and struggles to appropriately handle objects of varying sizes. Consequently, it is limited in capturing multidimensional information based

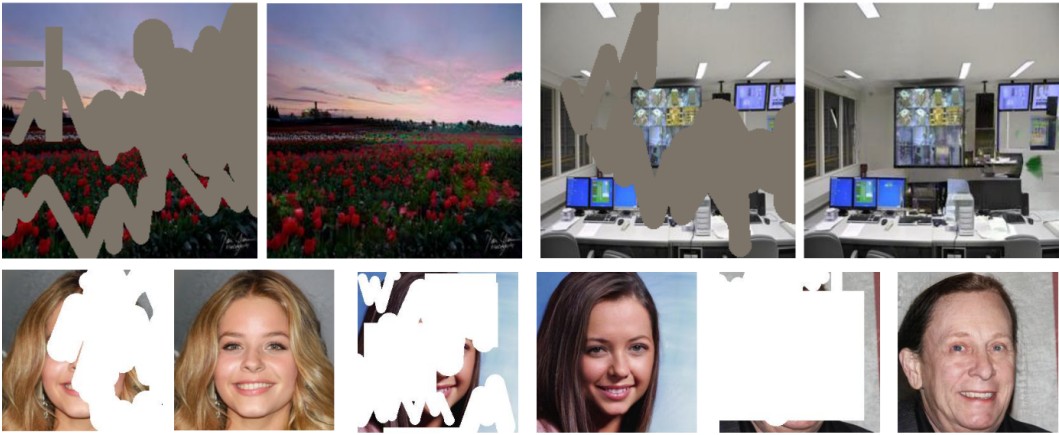

Figure 1: Successfully restored inpainting results by the proposed Multi-Scale Window based Transformer Network (MSWTN). In each image pair, the left side depicts the masked input image, while the right side shows the restored image.

on object sizes, resulting in restricted contextual information (Ren et al., 2022). Therefore, we propose a Multi-Scale Window based Transformer Network (MSWTN), capable of restoring even large missing regions. MSWTN incorporates the Multi-Scale Window-based Transformer (MSWT) block to perform parallel self-attention using multi-scale windows. This approach aims to acquire more diverse contextual information by employing various windows simultaneously. The features processed by self-attention in parallel are merged using the Multi-Scale Window-based Polarized Self-Attention (MW-PSA) mechanism to achieve effective fusion. MW-PSA combines channel and spatial information, providing better results compared to traditional attention mechanisms. Moreover, to reduce computational cost and enhance non-local interactions, the model utilizes multi-head contextual attention (MCA) (Li et al., 2022) instead of traditional multi-head self-attention. MCA conducts self-attention only using valid tokens based on a binary mask that distinguishes between the missing and normal areas in images, calculating non-local information. Valid tokens are determined by this binary mask, which is updated at each transformer block, gradually increasing the count of valid tokens. In this paper, we propose a Selective Mask Update to reflect key information from the features that undergo self-attention.

The experimental results, as shown in Figure 1, successfully fill in large missing regions. Furthermore, we also demonstrate superior performance on the respective dataset compared to other image inpainting models. The advantages of the method proposed in this paper are as follows:

- In this study, we propose MSWTN, an image inpainting model designed to restore large missing areas with high quality. This model utilizes a multi-scale window-based transformer architecture and outperforms existing inpainting models.
- MSWTN consists of the MSWT module, which conducts parallel self-attention using windows of multiple dimensions. This allows the model to capture various contextual information for restoration. We also propose the MW-PSA module to effectively fuse features that underwent MCA in MSWT.
- To achieve efficient mask updating, we proposed the Selective Mask Update module. The module extracts tokens from the output of MSWT and updates the mask by incorporating the positions of these tokens. Consequently, MSWTN efficiently fills the missing regions by prioritizing tokens that contain key information through self-attention.

## 2 RELATED WORK

### 2.1 IMAGE INPAINTING

Image inpainting, also known as image completion, is the process of filling large missing regions in images with plausible content. It holds significant interest in the field of computer vision. Deep

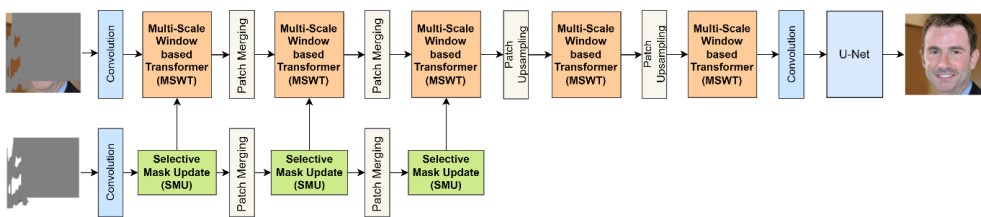

Figure 2: The proposed MSWTN in this paper. It consists of the MSWT module for image restoration and the SMU module for mask updating. The input image and mask pass through convolutional neural network layers, and the final output is restored using convolutional neural network layers

learning has had a profound impact on image inpainting and has facilitated advancements in this field. Methods based on CNNs include encoder-decoder networks that learn the relationships between missing and non-missing areas in order to restore them (Yu et al., 2019). However, architectures that solely rely on CNNs have limitations when it comes to filling large missing areas. To address this, attention-based modules were integrated into the existing encoder-decoder structures to achieve more photorealistic results for the missing areas (Liu et al., 2019; Ronneberger et al., 2015; Yu et al., 2021). Moreover, in pursuit of effectively filling even larger missing regions, models using multiple transformer blocks were introduced (Li et al., 2022; Yan et al., 2018) to comprehend long-range semantics for the missing areas. In (Yan et al., 2018), different patch sizes were processed for each attention head, enabling the capture of context information from various patch sizes. However, even with these models, limitations persist when dealing with challenging datasets due to the insufficient contextual information for larger missing areas.

## 2.2 VISION TRANSFORMER

In the field of computer vision, transformers have gained attention for their successful performance. The first vision transformer designed for images is ViT (Dosovitskiy et al., 2021), which divides images into fixed-size patches for computation. However, performing self-attention on the entire image using fixed-size patches results in high computational costs. To address this, the Swin Transformer (Liu et al., 2021b) introduced Window-based Multi-head Self-Attention (W-MSA), where self-attention is conducted only within specific windows. However, due to the fixed window sizes within transformer blocks, it is not possible to fully consider the impact of different window sizes. According to previous research (Li et al., 2022; Wang et al., 2004), fixed window sizes can limit model performance and restrict the availability of contextual information. Consequently, several multi-dimensional information-based vision transformer models (Chen et al., 2021; Li et al., 2022; 2021; Wang et al., 2004) have been proposed. In Li et al. (2022), patches divided by multi-dimensional windows within a transformer block are individually subjected to self-attention. In Chen et al. (2021), the image patches are divided into two branches based on their sizes in order to extract multi-dimensional feature maps for cross-attention. These models have demonstrated better performance than traditional vision transformer models. In this paper, we introduce MSWTN for high-quality image inpainting, which is using multi-dimensional windows to obtain context information without constraints imposed by window sizes.

## 3 METHOD

### 3.1 OVERALL MODEL

The proposed MSWTN architecture is depicted in Figure 2. It takes a masked image and its corresponding binary mask as inputs, with the goal of restoring the missing areas. The model consists of two main blocks: the MSWT (Multi-scale Window-based Transformer) block responsible for image restoration, and the SMU (Selective Mask Update) block responsible for updating the mask. The updated mask is then used as an input to MSWT, performing MCA (Multi-Head Contextual Attention) (Li et al., 2022) along with the input image. In MCA, the updated mask allows for self-attention among valid tokens. The input image undergoes tokenization through a convolution block and then

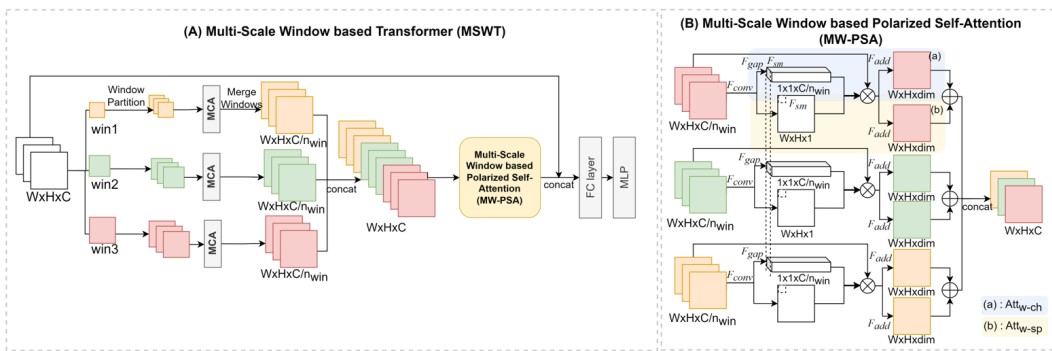

Figure 3: The proposed MSWT module, the core component of the proposed model. (A) depicts the entire network structure of MSWT, while (B) illustrates the MW-PSA network designed for the fusion of features that have passed through MCA.

passes through five transformer blocks. The initial convolution block is used for two main purposes: firstly, utilizing local inductive priors in the initial layers can result in improved representations, and secondly, downsampling the original image reduces computational complexity. The input dimensions of the five transformer blocks vary due to patch merging and patch upsampling. Through patch merging, the width and height dimensions of features in each layer decrease by half, while the channel dimension doubles. Conversely, patch upsampling is calculated in reverse compared to patch merging. The output of the last transformer block is passed through convolutional neural network layers to restore the original input size. Furthermore, U-Net (Ronneberger et al., 2015) is employed at the end to enhance the local textures, thereby improving the fine details of the image.

## 3.2 Multi-Scale Window based Transformer

In this study, we propose a new transformer block called MSWT that enhances the restoration performance of large missing regions. Unlike conventional vision transformer structures, MSWT uses multi-scale windows to perform window partitioning for window-based self-attention. This approach aims to capture and utilize information based on multiscale windows. As shown in Figure2 (A), each transformer block consists of four main modules: MCA (Multi-head Contextual Attention), MW-PSA (Multi-Scale Window-based Polarized Self-Attention), FC (Fully Connected) layer, and MLP (Multi-Layer Perceptron). This can be formulated as follows:

$$
\begin{aligned}
x^{l+1} &= \text{MLP}(\text{FC}(\acute{x}^l)) \\
\acute{x}^l &= \text{concat}(\text{MW} - \text{PSA}(\hat{x}^l), x^l) \\
\hat{x}^l &= \text{concat}(\{\text{MCA}_i(x_i^l)\})
\end{aligned}
\tag{1}
$$

In equation (1), the function $\text{concat}()$ is feature concatenation. Where the input feature denoted by $x \in \mathbb{R}^{H \times W \times C}$ and the number of windows $n_{win}$, $x$ is divided into $i$-th window in $l$-th block. After passing through MCA in parallel, the features are restored to dimension $H \times W \times \frac{C}{n_{win}}$ and then concatenated along the channel dimension to output $\hat{x}^l$. $\hat{x}^l$ is fused into a single output through $\text{MW} - \text{PSA}$, and the output is combined with initial input $x^l$. This passes through the FC layer and MLP to produce the final feature $x^{l+1}$ of the block. In this paper, number of windows is $n_{win} = 3$ and each window's size set to $win_{i(i=1,2,3)} = \{4, 8, 16\}$. We determined them through performance comparisons based on experiments.

### 3.2.1 Multi-head Contextual Attention (MCA)

In image inpainting, a masked image consisting of valid tokens and invalid tokens is often used. Using existing window based self-attention leads to inefficiencies in computational cost since it operates on both valid and invalid token. Moreover, it can weaken the information of valid tokens, limiting its effectiveness in filling the missing pixel information. To address this, we introduce

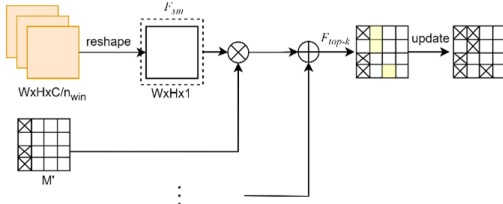

Figure 4: The visualization of proposed Selective Mask Update.

the Multi-Head Contextual Attention (MCA) (Li et al., 2022) to our model, which performs self-attention only on valid tokens based on the mask. This achieve higher quality inpainting results than results from normal self-attention. The equation for MCA is as follows:

$$\text{Att}(\mathbf{Q}, \mathbf{K}, \mathbf{V}) = \text{Softmax}(\frac{\mathbf{Q}\mathbf{V}^T + \mathbf{M}'}{\sqrt{d_k}})\mathbf{V} \tag{2}$$

where $\mathbf{Q}, \mathbf{K}, \mathbf{V}$ is query, key, value respectively, and $d_k$ is a scaling factor. $\mathbf{M}'$ stands for the mask, where it is set to 0 for valid tokens (non-missing areas) and -3000 for invalid tokens(missing areas). Therefore, MCA determines self-attention values based on whether the token is valid of not.

### 3.2.2 MULTI-SCALE WINDOW BASED POLARIZED SELF-ATTENTION (MW-PSA)

The output of MCA passes through MW-PSA for effective fusion. MA-PSA is devised based on the Polarized Self-Attention (Liu et al., 2021a), and its structure is shown in Figure 2 (B). The input to MW-PSA is denoted as $\hat{x}^l \in \mathbb{R}^{H \times W \times C}$ , and the results from MCA for each window size are denoted as $\hat{x}_i^l \in \mathbb{R}^{H \times W \times \frac{C}{n_{win}}}$ , then $\hat{x}^l = \text{concat}(\{\hat{x}_i^l, i = (1, \cdots, n_{win})\})$ in this context. The equation of MW-PSA is as follow:

$$\text{MW} - \text{PSA}(\hat{x}^l) = \text{Att}_{w-ch}(\hat{x}^l) + Att_{w-sp}(\hat{x}^l) \tag{3}$$

Where $\text{Att}_{w-ch}(\cdot)$ is window-based channel-only attention, $\text{Att}_{w-sp}(\cdot)$ is also window-based spatial-only attention. The final output of MW-PSA is obtained by combining the features that have passed through both attention modules. The equation of $\text{Att}_{w-ch}(\cdot)$ is as follow:

$$
\begin{aligned}
\text{Att}_{w-ch}(\hat{x}^l) &= \text{concat}(\{F_{add}(\hat{x}_i^l \odot_{ch} \hat{w}_i^l), \\
& i = (1, \cdots, n_{win})\}), \\
\hat{w}^l &= F_{sm}(\text{concat}(\{w_i^l, i = (1, \cdots, n_{win})\})), \\
\hat{w}_i^l &\in \mathbb{R}^{1 \times 1 \times \frac{C}{n_{win}}}, \\
w_i^l &= F_{gap}(F_{conv}(\hat{x}_i^l)) \in \mathbb{R}^{1 \times 1 \times \frac{C}{n_{win}}}
\end{aligned}
\tag{4}
$$

$\hat{x}_i^l$ is passing through 1x1 convolution layer $F_{conv}$ and global average pooling $F_{gap}$ , and results in attention weight $\hat{w}_i^l$. The attention weight for each window features $\hat{w}_i^l$ are concatenated and then passed through softmax operation $F_{sm}$ along the channel dimension. The weights for chnnel-only attention $\hat{w}_i^l$ corresponding i-th window feature are computed through channel-wise multiplication with the input $\hat{x}_i^l$ . Subsequently, element-wise addition $F_{add}$ is performed, resulting in a dimension $H \times W \times 1$. The outputs of $F_{add}$ are concatenated to have same dimension as the initial one. The same process applies to window-based spatial-only attention, with the attention weights' dimension is transformed to $\hat{w}_i^l \in \mathbb{R}^{H \times W \times 1}$. Through this process, it's possible to perform spatial and channel attention not just for a single feature but for multiple window features. This essentially allows conducting both attention for various window-based results, which is the structure ensures that minimal information leak occurs while enabling the fusion of MCA outputs.

### 3.2.3 SELECTIVE MASK UPDATE

The mask is binary, where valid tokens have a value of 1, while invalid tokens are assigned a value of 0. This mask determines whether the MCA operation is applied to the position of the respective

token. The goal of SMU is to fill all the empty regions of the image, making the mask update process a transformation of all tokens into valid tokens in the mask. The input consists of the window features $\hat{x}_i^l$ after MCA and the inverse mask $M'$ to be updated. where $m'_j$ signifies the tokens within the inverse mask $M' = \{m'_j | j = 0, 1, \cdots, s\}$ , and $s$ represents the number of tokens. The mask updating process proceeds as follows:

$$A = F_{\text{top}-\text{k}}(\sum_{i=1}^{n_{win}} F_{sm}(\sigma(\hat{x}_i^l)) \odot M')$$ (5)

$\hat{x}_i^l$ is adjusted in dimension through reshape $\sigma$ to make the one channel, followed by softmax operation across pixels $F_{sm}$. This process assigns importance to each pixel's location, determining the significance based on the pixel's positional information. We conduct element-wise multiplication between the positional information and inverse mask to determine which positions of tokens in the missing regions. This operation is performed for all window-based information and the and results are summed up. The combined features then undergo Top-K selection $F_{\text{top}-\text{k}}$ , K is adaptable based on the image size. Consequently, the Top-K Selection operation, the set of indices token A corresponding to the K most significant tokens is generated as output. The tokens corresponding to these indices in set A are updated as valid tokens, completing the final mask update process.

$$m'_j = \left\{ \begin{array}{l} 0, \text{if} j \in A \\ 1, \text{if} j \notin A \end{array} \right\}$$ (6)

Therefore, the Selective Mask Update introduced in this paper utilizes the feature information after MCA to update the mask. Through Top-k selection, it extracts essential information and updates the positions of this information. This approach allows the utilization of crucial information within the image to restore empty regions, resulting in improved outcomes.

## 3.3 LOSS FUNCTIONS

To enhance the quality of image generation, this paper combines adversarial loss (Creswell et al., 2018), perceptual loss (Johnson et al., 2016), and style loss (Gatys et al., 2016) in an optimal manner to train the model.

**Adversarial Loss.** In order to train the Generative Adversarial Network (GAN) stably, which is widely employed in image generation, this paper introduces the non-saturating adversarial loss (Creswell et al., 2018).

$$\mathcal{L}_G = -E_{\hat{x}} [log (D (\hat{x}))]$$
$$\mathcal{L}_D = -E_x [log (D (x))] - E_{\hat{x}} [log (1 - D (\hat{x}))]$$ (7)

In the equation (7), $x$ represents real images, and $\hat{x}$ represents fake images generated by the generator. Both the generator and discriminator are optimized according to their respective loss functions.

**Perceptual Loss.** The perceptual loss is obtained by using feature maps from a pre-trained VGG-16 network. It involves comparing the feature maps of the generated images with the feature maps of the real images, aiming to create more realistic images by minimizing the differences between these feature maps.

$$\mathcal{L}_{perc} = \sum_{i=1}^{N} \eta_i \|\phi_i(\hat{x}) - \phi_i(x)\|_1$$ (8)

Where $\phi_i$ is the activation function of i-th layer in the pretrained VGG-16 network, and $\eta_i$ is the scailing coefficient.

**Style Loss.** For achieving finer texture restoration, Style loss has been incorporated. Similar to the perceptual loss, Style loss also employs the feature maps of pre-trained VGG-16 layers. The Gram matrices of these feature maps and the ground truth (GT) are computed, and then the Mean Squared Error (MSE) is calculated between these Gram matrices. The process can be represented by the following equation:

$$\mathcal{L}_{style} = \sum_{i=1}^{N} \eta_i \left\| \phi_i(\hat{x})(\phi_i(\hat{x}))^{\text{T}} - \phi_i(x)(\phi_i(x))^{\text{T}} \right\|_1$$ (9)

Table 1: Quantitative comparison for the Places365 and CelebA-HQ datasets. All images are of size 256 x 256, and the results are divided between Small Mask and Large Mask. The highest performing values are indicated in bold.

| Datasets | | Places368-standard | | CelebA-HQ | |
|---|---|---|---|---|---|
| Mask size | | Small | Large | Small | Large |
| FID | MSWTN(ours) | **1.07** | **2.74** | **2.61** | **4.85** |
| | MAT[11] | 1.15 | 2.99 | 2.94 | 5.16 |
| | CoModGAN[18] | 2.06 | 6.18 | 5.12 | 14.56 |
| | DeepFill v2[19] | 6.83 | 22.23 | 5.14 | 12.79 |
| | MADF[20] | 12.80 | 18.42 | 5.55 | 11.13 |
| | RFR[21] | 8.34 | 25.88 | 8.75 | 23.81 |
| P-IDS(%) | MSWTN(ours) | **19.08** | 12.30 | **20.48** | **14.17** |
| | MAT[11] | 18.89 | 13.33 | 10.56 | 13.90 |
| | CoModGAN[18] | 18.98 | **16.79** | 5.45 | 0.43 |
| | DeepFill v2[19] | 4.39 | 1.08 | 6.68 | 1.30 |
| | MADF[20] | 14.39 | 8.45 | 3.74 | 0.80 |
| | RFR[21] | 13.19 | 3.90 | 0.05 | 0.03 |
| U-IDS(%) | MSWTN(ours) | **39.29** | **31.63** | **32.58** | 24.47 |
| | MAT[11] | 37.62 | 30.23 | 22.85 | **25.13** |
| | CoModGAN[18] | 35.12 | 24.43 | 15.87 | 13.80 |
| | DeepFill v2[19] | 22.58 | 10.87 | 17.67 | 3.07 |
| | MADF[20] | 18.78 | 16.33 | 15.84 | 3.17 |
| | RFR[21] | 20.61 | 9.36 | 12.93 | 0.84 |

**Overall Loss.** In the overall loss function, $R_1$ regularization (Mescheder et al., 2018) has been applied, and the equation for this regularization term is as follow.

$$R_1 = E_x \parallel \nabla D(x) \parallel \tag{10}$$

The final overall loss function for the generator is composed as follows:

$$\mathcal{L} = \mathcal{L}_G + \lambda_1 R_1 + \lambda_2 \mathcal{L}_{perc} + \lambda_3 \mathcal{L}_{style} \tag{11}$$

Where $\lambda_1 = 10, \lambda_2 = 0.1, \lambda_3 = 0.1$

## 4 EXPERIMENTS

### 4.1 DATASETS AND IMPLEMENTATION DETAILS

We conducted experiments using two datasets, namely Places365 (Zhou et al., 2018) and CelebA-HQ (Karras et al., 2018). The image resolution for both datasets was set at 256x256 pixels. For the Places365 dataset, approximately 1.8 million images were used for training, and 36,500 images were used for validation. In the case of CelebA-HQ, the training set consisted of 27,007 images, and the validation set comprised 2,993 images. Two types of masks, referred to as small and large masks, were employed for testing. These masks were generated using the free-form mask generation approach from DeepFill v2 (Yu et al., 2019). For the small mask, we defined the range of the missing area ratio as [0.0, 0.4], and for the large mask, we set it to [0.0, 0.9]. Masks were generated through random sampling within these specified ranges. These mask ratios were determined with Li et al. (2022).

We used a batch size of 16 and employed the Adam optimizer. The learning rate was set to $1x10^{-3}$, $\beta_1 = 0$ , $\beta_2 = 0.99$. The experiments were conducted on a PC equipped with two parallel NVIDIA GeForce RTX 3090 GPUs.

### 4.2 RESULTS

We compared the results of our experiments with those of state-of-the-art in image inpainting. The comparison is carried out through both qualitative and quantitative analyses. In the quantitative

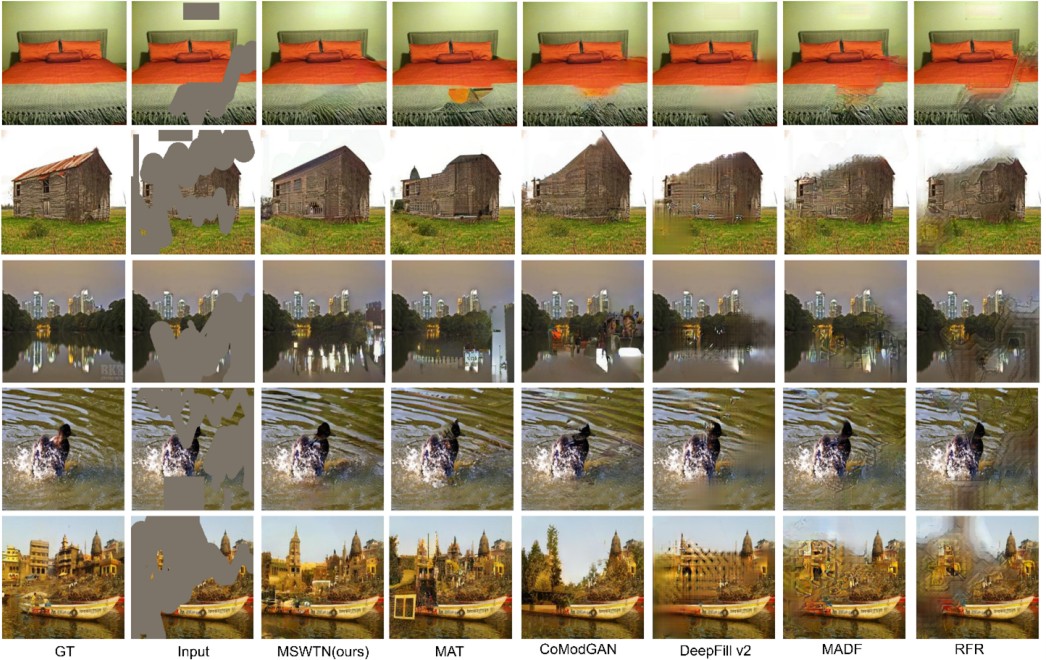

Figure 5: Qualitative comparison between the proposed model and other existing models for the Places365 dataset. From left to right: Ground Truth (GT), Masked Input, Proposed Model, and results from comparative models.

comparison, three metrics—FID (Heusel et al., 2017), P-IDS (Zhao et al., 2021), and U-IDS (Zhang et al., 2018a)—are utilized for comparison. PSNR and SSIM (Wang et al., 2004) are excluded from our evaluation as they have been shown in numerous studies (Ledig et al., 2017) to exhibit significant discrepancies with human perception when assessing the quality of high-resolution images.

### 4.2.1 COMPARISON WITH SATE OF THE ARTS

**Quantitative comparisons.** In this experiment, we compared our proposed model MSWTN with several state-of-the-art (SOTA). To ensure a fair comparison, both training and testing were conducted on the same dataset, and the same set of masks was used for testing across all models. The models to compared are MAT (Li et al., 2022), CoModGAN (Zhao et al., 2021), DeepFillv2 (Yu et al., 2019), MADEF (Zhu et al., 2021), and RFR (Li et al., 2020). Quantitative comparison results presented in Table 1 indicate that our proposed MSWTN outperforms the comparison models. When evaluating on the Places dataset using small masks, MSWTN achieves the highest performance across all metrics. For large masks, MSWTN achieves the highest performance in all metrics except for PIDS. Similar trends are observed on the CelebA-HQ dataset, where MSWTN achieves the best performance for small masks and also best performance for large masks, excluding UIDS. The CoModGAN achieves the highest PIDS score on large masks for the Places dataset, while the MAT records the highest UIDS score on large masks for the CelebA-HQ dataset. Our proposed MSWTN ranks third in terms of PIDS and second in UIDS scores for these datasets.

**Qualitative comparisons.** For visual comparison of the results, we compare the test results of our proposed MSWTN model and five other comparative models with the ground truth images from the CelebA-HQ and Places365 datasets. Figure 5 shows the test results of the proposed MSWTN model and five comparative models for the Places365 dataset. The results are all of size 256x256 and restoration with large masks. It is evident that the result images from MSWTN closely resemble the ground truth (GT) and exhibit fewer artifacts compared to the results of other models.

Table 2: Quantitative comparison for the ablation study. The dataset used is CelebA-HQ, and Models A, B, and C represent the models with specific modules removed from the Full Model.

| Model | FID | P-IDS(%) | U-IDS(%) |
|---|---|---|---|
| Full Model | **4.85** | **14.17** | **24.47** |
| A - MSWT | 5.20 | 12.13 | 21.23 |
| B - MW-PSA | 5.01 | 11.89 | 23.94 |
| C - SMU | 4.99 | 12.60 | 23.67 |

### 4.2.2 ABLATION STUDY

In this section, an experiment was conducted to assess the impact of the modules that constitute the proposed model on performance. The experiment was carried out using the CelebA-HQ dataset under the same conditions as mentioned in section 4.1. The goal was to compare performance by gradually removing modules from the full model proposed in this paper. The quantitative comparison of these results can be observed in Regarding performance, all three metrics were best achieved when the full model was used. In case A, the performance is demonstrated when the MSWT module is removed, and instead, the basic Swin-Transformer is utilized. Across all three metrics, a degradation of over 10% compared to the full model is evident. This indicates that the proposed MSWTN is significantly influenced by the MSWT module, implying that the structure incorporating the multi-scale window proposed is effective for inpainting tasks. In case B, the result is obtained when the MW-PSA is omitted from the full model, and the output of MSWT is fused with a simple convolutional neural network layer. In this case, the performance demonstrates the most significant drop compared to other models. Therefore, the fusion process of MSWT's output is found to be highly crucial. Lastly, case C represents the results when only the Selective Mask Update is omitted from the Full Model, and the conventional mask update mechanism is used. Among the three modules, this case shows the least performance discrepancy compared to the full model.

## 5 CONCLUSION

We propose a multi-scale windows-based transformer for high-quality video inpainting. We conducted parallel self-attention through multi-dimensional windows to incorporate diverse contextual information, achieving superior performance compared to existing models. Furthermore, we introduce a Multi-scale Window Polarized Self-attention(MW-PSA), which efficiently fuses features obtained through MCA in both channel and spatial dimensions. Additionally, addressing the inefficiency of existing mask update methods, we propose the Selective Mask Update (SMU) approach. This method involves updating the masks of prioritized regions based on tokens that have undergone self-attention. We have observed that our model exhibits superior performance compared to other models in both quantitative and qualitative evaluations.

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

APPENDIX

## A   MASK DETAILS

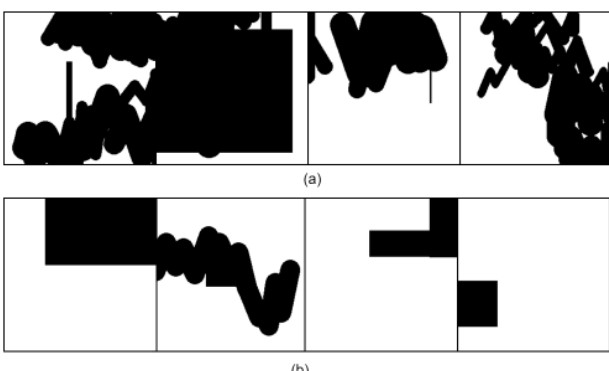

Figure 6: The images of the free-form masks used in the experiments, (a) represents a sample of the large mask, and (b) represents a sample of the small mask.

We randomly generated free-form masks. Free-form masks were created based on Yu et al. (2019), generating random-sized rectangles and specifying the ranges for the number of rectangles and strokes to construct both large masks and small masks. The stroke ranges for both large masks and small masks were referenced from Li et al. (2022). For large masks, the number of generated rectangles is uniformly sampled within [0, 5], and the number of strokes is randomly sampled from [0, 9]. For small masks, the number of generated rectangles is sampled from [0, 3], and the number of strokes is constrained within [0, 4]. The generated large masks were used for training on the experimental dataset, and for testing, both large masks and small masks were employed. Figure 6 shows each mask dataset.

## B   ADDITIONAL ABLATION STUDY

Table 3: Quantitative comparison for the additional ablation study. Model (A) represents the Full Model, Models (B), and (C) represent the models with specific modules removed from the Full Model.

| Model | LPIPS | PSNR | SSIM |
|---|---|---|---|
| (A) Full Model | **0.097** | **23.30** | **0.82** |
| (B) - MW-PSA | 0.102 | 22.89 | 0.81 |
| (C) - SMU | 0.100 | 23.16 | 0.81 |

In this section, we provide additional results for the ablation study. The ablation study was performed on the CelebA-HQ dataset, and Table 3 presents a quantitative comparison conducted as an additional performance evaluation to assess the impact of each module. As additional performance metrics, we used LPIPS (Zhang et al., 2018b), PSNR and SSIM (Wang et al., 2004). It was observed that the Full Model outperformed the others in all three metrics. Among these metrics, it was observed that model (B) exhibited the most significant difference from the Full Model in terms of LPIPS and PSNR. Specifically, the PSNR showed a clear difference compared to other metrics, and this observation also applied to model (C). In the case of SSIM, both models showed the least noticeable difference when compared to the Full Model. However, model (B) displayed the lowest performance in terms of LPIPS and PSNR compared to the Full Model, indicating the significant impact of the MW-PSA module.

Qualitative comparisons for this are presented in Figure 7. It can be observed that model (A) restored the image most closely to reality compared to the other models. Specifically, detailed areas such as the eyes were well-formed and did not appear artificial.

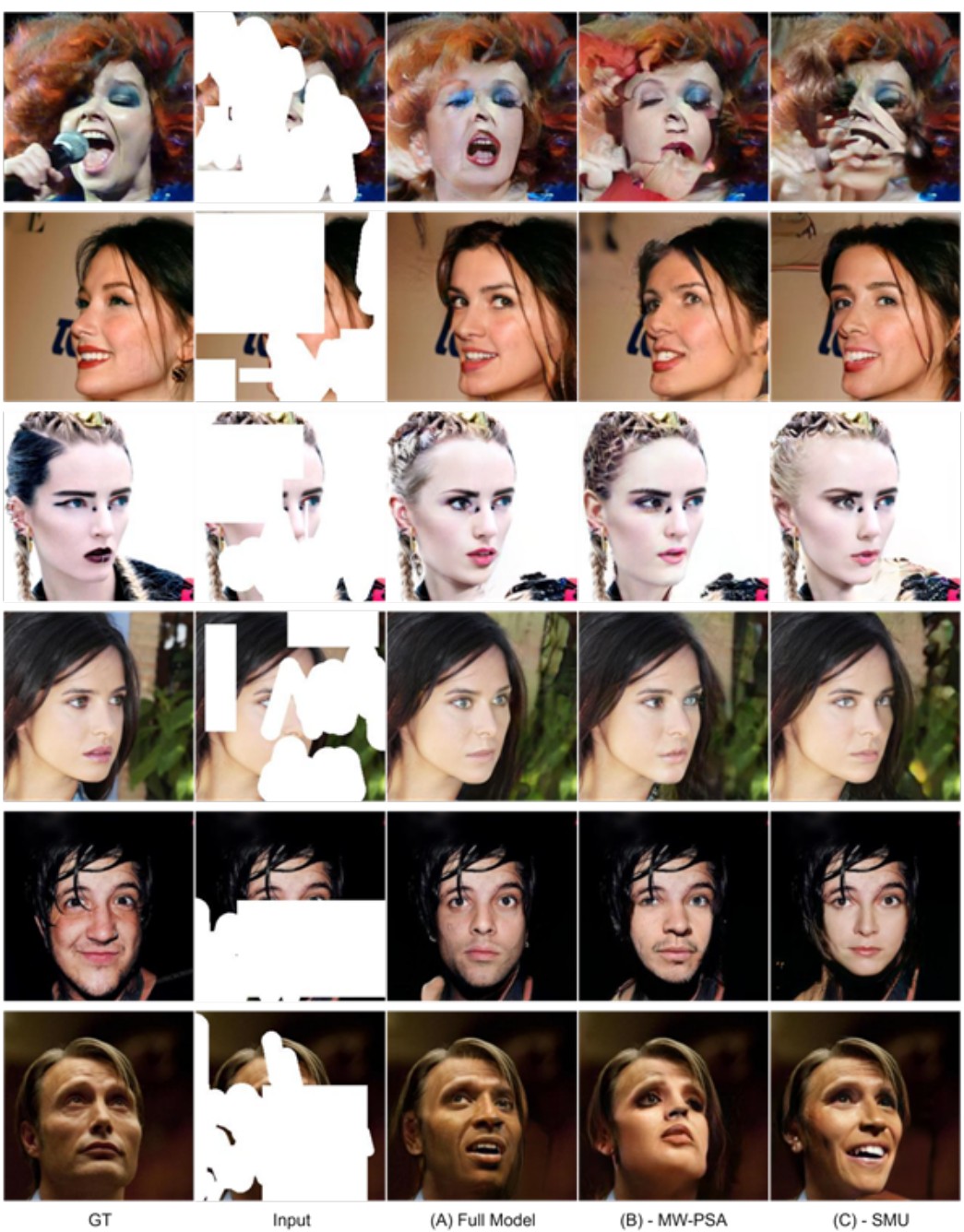

Figure 7: Qualitative comparison for the additional ablation study. (A), (B), and (C) models are the same as the models in Table 3.

## C  ADDITIONAL RESULTS

We conducted experiments on the CelebA-HQ (Karras et al., 2018) dataset and the Places365 (Zhou et al., 2018) dataset. In this regard, we provide additional Qualitative comparisons for experiments conducted with the proposed model MSWTN and various state-of-the-art methods. Figure 8 shows visual results for the CelebA-HQ dataset, while Figure 9 depicts additional visual results for the Places dataset. The proposed model MSWTN exhibits more realistic restoration results compared to other models, with less noise in the images.

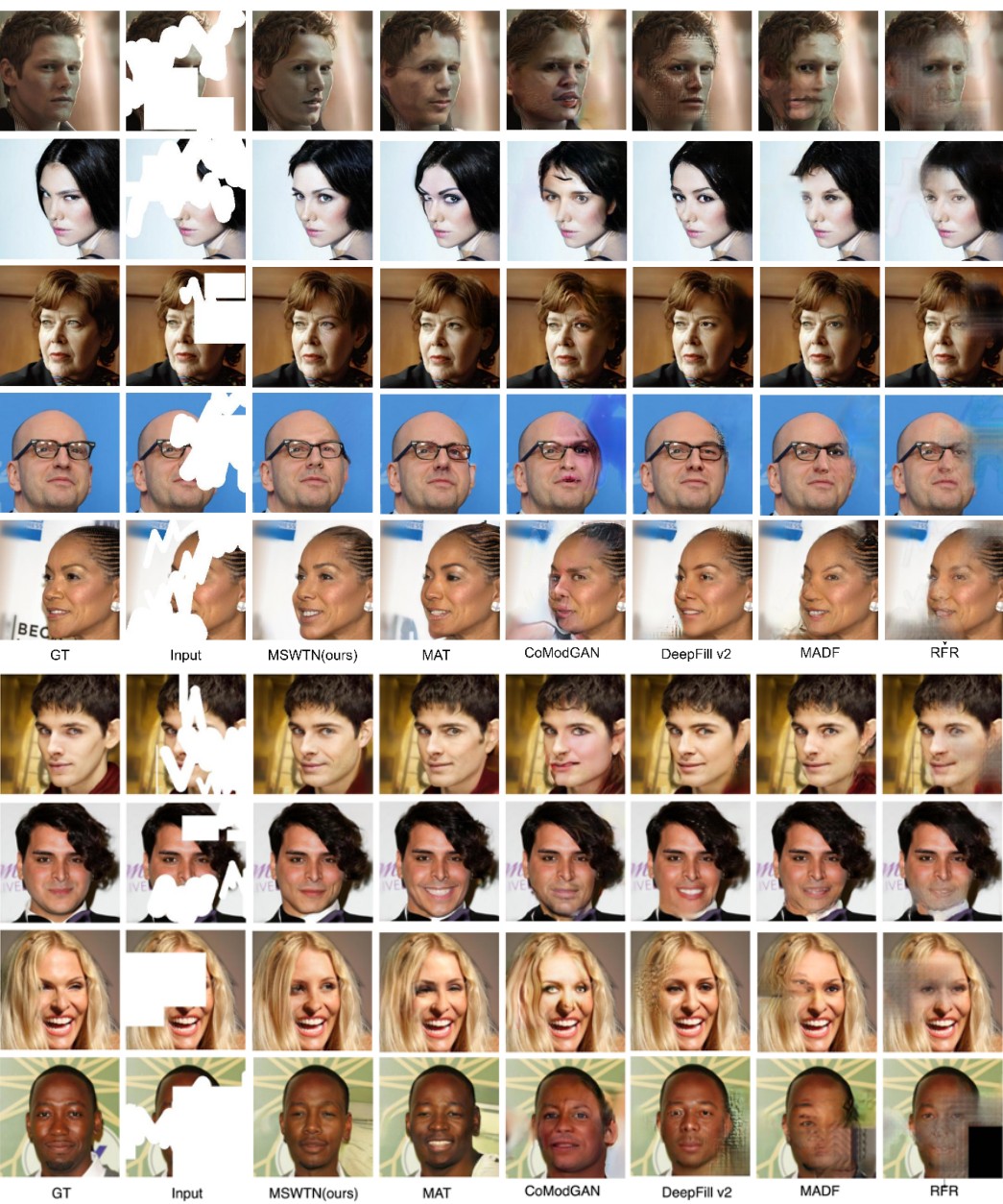

Figure 8: Qualitative comparison between the proposed model and other existing models for the CelebA-HQ dataset.

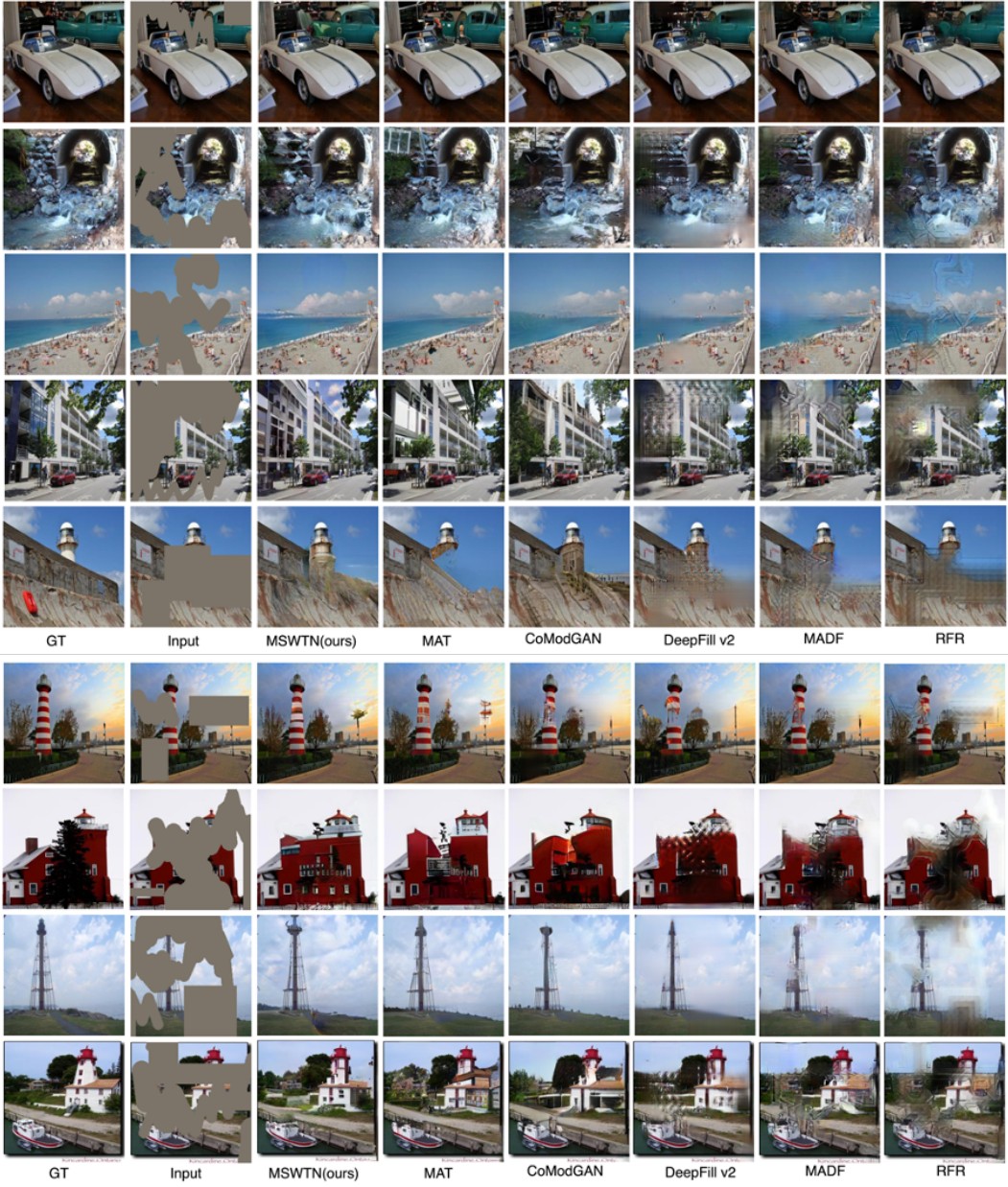

Figure 9: Qualitative comparison between the proposed model and other existing models for the Places365 dataset.

