# OpenReview forum: "Multi-Scale Window based Transformer Network for High Quality Image Inpainting"
_ICLR.cc/2024/Conference — Submitted to ICLR 2024_

### Official Review · Reviewer_UsMm · 2023-10-13

**Soundness:** 2 fair
**Presentation:** 2 fair
**Contribution:** 1 poor
**Rating:** 3
**Confidence:** 5

**Summary:**

This paper introduces a model for image inpainting. It identifies a gap in prior methods, noting that they often face challenges due to the lack of long-range dependencies and difficulty in capturing contextual information. To address these issues, the authors introduce a multi-scale window-based transformer designed to capture the effects of varying window sizes and gather essential contextual information. Additionally, they propose a selective mask update strategy that extracts crucial data from features processed by self-attention, leading to the generation of higher-quality results.

**Strengths:**

- The proposed framework outperforms the compared baselines in terms of FID.

- Qualitatively, the proposed model surpasses the compared baselines.

**Weaknesses:**

- Marginal improvement: As shown in Table 1, the proposed method offers only a slight enhancement compared to MAT.
- Limited baselines for comparison: The study does not compare with diffusion models, such as stable diffusion inpainting.
- Limited novelty: Some existing works share similar insights with the proposed multi-scale window and selective mask update.

**Questions:**

No

---

### Official Review · Reviewer_Z4FZ · 2023-10-22

**Soundness:** 2 fair
**Presentation:** 1 poor
**Contribution:** 1 poor
**Rating:** 3
**Confidence:** 5

**Summary:**

This paper proposed an image inpainting method based on the multi-scale window transformer. Moreover, Multi-Scale Window-based Polarized Self-Attention (MW-PSA) and multi-head contextual attention (MCA) are used to further improve the performance. The Selective Mask Update (SMU) is proposed to reflect critical information from the features that undergo self-attention. The proposed method enjoys good performance, but it suffers from obviously limited novelty.

**Strengths:**

1. The proposed method is easy to follow and achieve good performance in the comparison shown in the paper.

**Weaknesses:**

1. The main concern is the limited novelty. Both multi-scale training and Windows-based attention are widely used in computer vision. So the combination of them is naive and intuitive to achieve slightly better performance. However, the authors did not provide more in-depth discussions to make their claims clearer. For example, what is the key difference between multi-scale+window-based attention and multi-scale dilated convolution used in AOT-GAN[1]?
2. Other techniques, such as MW-PSA and MCA, were all proposed in previous works. For the SMU, the authors miss enough discussions and clarifies about the other works of adaptive mask adjusting, such as Partial Conv[2], Gated Convolution[3], and so on.
2. The writing of this paper is non-standard. The egregious citation issues are evident throughout the entire document, making it difficult to establish the credibility and reliability of the sources used. 1) Incomplete citations: a substantial portion of the references lack crucial information such as the name of the original publisher and the journal name.  2) Incorrect publication years: the paper "High-fidelity pluralistic image completion with transformers" was published in CVPR2021, but the authors marked that it was published in 2018 (without any publisher information). 3) Wrong citations: the authors said "For these reasons, recent advancements in image inpainting have led to the proposal of Transformer-based models (Yan et al., 2018; Zhang et al., 2018a)", but both (Yan et al., 2018) and  (Zhang et al., 2018a) are completely irrelevant to attention or transformer.

[1] Zeng Y, Fu J, Chao H, et al. Aggregated contextual transformations for high-resolution image inpainting[J]. IEEE Transactions on Visualization and Computer Graphics, 2022.

[2] Liu G, Reda F A, Shih K J, et al. Image inpainting for irregular holes using partial convolutions[C]//Proceedings of the European conference on computer vision (ECCV). 2018: 85-100.

[3] Yu J, Lin Z, Yang J, et al. Free-form image inpainting with gated convolution[C]//Proceedings of the IEEE/CVF international conference on computer vision. 2019: 4471-4480.

**Questions:**

It is strongly recommended that the authors revise and correct the citation and referencing issues. Furthermore, a comprehensive review of the paper's content for factual accuracy and relevance to the topic should be considered to ensure the overall quality of the research.

---

### Official Review · Reviewer_NdAo · 2023-10-30

**Soundness:** 3 good
**Presentation:** 2 fair
**Contribution:** 2 fair
**Rating:** 5
**Confidence:** 5

**Summary:**

The paper introduces a Multi-Scale Window-based Transformer model dedicated to achieving high-quality image inpainting. This model is devised to effectively comprehend contextual information by incorporating a transformer network featuring multi-scale windows. To enhance this model's capabilities, a modified polarized self-attention network is employed to align with multi-window scales, optimizing the assimilation of significant contextual information. Moreover, the Selective Mask Update method is proposed to capture essential information from the output features, facilitating timely mask region updates, thus enhancing the utility efficiency of valid data. Experiments show that the proposed model generates superior results compared to other models on the benchmark dataset.

**Strengths:**

Strength:
1.The paper robustly explicates the design rationale behind the Multi-Scale Window-based Polarized Self-Attention (MW-PSA) mechanism. It effectively validates the application's effectiveness through experiments in large mask inpainting tasks. This marks the first successful implementation of such technologies in inpainting tasks.
2.The paper introduces an effective strategy for updating valid tokens within masks, which aids in enhancing task efficiency and the quality of generated results.
3.The presentation of this paper is professional and fluent. It has almost no expression errors and clearly elucidates the authors' contributions.

**Weaknesses:**

Weaknesses:
1. The complete omission of PSNR-oriented metrics in the comparative evaluation is an unreasonable approach. Typically, generative models tend to perform relatively poorly on PSNR-oriented metrics, and the exclusion of discussion around these metrics is perplexing. Transformer-based methods often excel, and for some downstream tasks in inpainting services, the completion results should demonstrate a certain fidelity. To comprehensively showcase the algorithm's performance, it is imperative to include experiments addressing these metrics.

2. The conducted ablation experiments are exceedingly inadequate, providing only qualitative discussions. The scale, details, and ablation experiments related to the U-Net network settings should be included at the module level within the processing pipeline. Crucial details regarding hyperparameters, such as the k value in the Top-k selection of the Selective Mask Update (SMU) approach, merit comprehensive ablation studies.

3.The motivation behind the 'strengthening local textures and enhancing fine details of the images' by simply concatenating U-Net networks is both perplexing and unconvincing. In practice, regression-based CNN methods of this nature are often considered to be the cause of over-smoothing and detail loss.

4.The novelty and innovativeness of the paper are relatively low. Similar designs employing a Multi-Scale Window-based Transformer have been proposed in some articles. Although the algorithm's application tasks and design purposes might differ, the authors have neglected a thorough literature review, completely omitting references to highly similar works such as [1] and [2] . Furthermore, the MW-PSA appears to be a simple modification of the existing PSA under the input mechanism of parallel multi-windows, while the MCA is directly borrowed without substantial innovation.
[1] Ren, P., Li, C., Wang, G., Xiao, Y., Du, Q., Liang, X., & Chang, X. (2022). Beyond fixation: Dynamic window visual transformer. In Proceedings of the IEEE/CVF Conference on Computer Vision and Pattern Recognition (pp. 11987-11997).
[2] Cheng, R., Zhuang, Z., Zhuang, S., Xie, L., & Guo, J. (2023). MSW-Transformer: Multi-Scale Shifted Windows Transformer Networks for 12-Lead ECG Classification. arXiv preprint arXiv:2306.12098.

**Questions:**

See weaknesses.

---

### Official Review · Reviewer_8FYW · 2023-10-31

**Soundness:** 3 good
**Presentation:** 3 good
**Contribution:** 2 fair
**Rating:** 3
**Confidence:** 5

**Summary:**

This paper propose a multi-scale transformer architecture for image inpainting and a Selective Mask Update method for mask updating. Experiments show the effectivenss of the proposed method with a resolution of 256.

**Strengths:**

1.The proposed parallel self-attention through multi-dimensional windows to incorporate diverse contextual information, achieving superior performance compared to existing models.
2. MW-PSA efficiently fuses features obtained through MCA in both channel and spatial dimensions.
3. The experiments show the superiority compared to other models in both quantitative and qualitative evaluations.

**Weaknesses:**

1. This paper aims for high quality image inpainting, but it changed to high-quality video inpainting in conclusion. It is a resubmitted paper? The author should check carefully before submission.
2. In my opinion, "high-quality: means higher quality images with higher resoultions, but in this paper, all experiments are only conducted on 256x256. The author should add more experiments on higher resolution to show the effectiveness of their method. Besides, since some methods like CoModGAN are trained on 512x512, it is unfair to compared with them on 256x256.
3. Why not compare with diffusion models based methods which show superiority on large missing areas over GAN based methods? The experiments of this paper cannot convince me.
4. The writing of this paper needs improving:
"Places368-standard" in table 1
The mask color is different between row 1 and row2 in Figure 1

**Questions:**

See Weaknesses.

---

### Meta-Review · Area_Chair_cz7q · 2023-12-06

**Metareview:**

This paper proposes a Transformer architecture that incorporates multiple window sizes to aggregate context information. The paper demosntrates competitive performance to existing work with improved quantitative and qualitative results. However, the novelty and the proposed technique has been proposed by prior works. The reveiewers provided detailed evidence backing this. Additionally, the presentation and writing of the paper needs to be improved. For example, the conclusion does not even match the paper, stating that they proposed a method "video inpainting".

**Justification For Why Not Higher Score:**

The paper has limited novelty and the writing isn't ready for publication.

**Justification For Why Not Lower Score:**

N/A

---

### Decision · Program_Chairs · 2024-01-16

Reject